Impact of agricultural management on bacterial laccase-encoding genes with possible implications for soil carbon storage in semi-arid Mediterranean olive farming

Moreno Beatriz
http://orcid.org/0000-0002-8435-066X Benitez Emilio emilio.benitez@eez.csic.es
Department of Environmental Protection, CSIC-Estacion Experimental del Zaidin (EEZ) , Granada , Spain
Johnson Stephen
Electronic publication date: 2016 Jul 21
Publication date: 2016
Volume: 4
Electronic Location ID: e2257
Received 2016 Mar 16; Accepted 2016 Jun 24
Copyright: © 2016 Moreno & Benitez
Copyright year: 2016
Copyright holder: Moreno & Benitez
License: This is an open access article distributed under the terms of the Creative Commons Attribution License, which permits unrestricted use, distribution, reproduction and adaptation in any medium and for any purpose provided that it is properly attributed. For attribution, the original author(s), title, publication source (PeerJ) and either DOI or URL of the article must be cited.
License URL: https://creativecommons.org/licenses/by/4.0/

Keywords: C sequestration, Cover crops, Olive farming, Litter decomposition, Bacterial laccase, Humic acids

Funding: ERDF-cofinanced CGL2009-07907 The work was supported by ERDF-cofinanced grant CGL2009-07907 from the Spanish Ministry of Economy and Competitiveness. The funders had no role in study design, data collection and analysis, decision to publish, or preparation of the manuscript.

==============================
Background: In this work, we aimed to gain insights into the contribution of soil bacteria to carbon sequestration in Mediterranean habitats. In particular, we aimed to use bacterial laccase-encoding genes as molecular markers for soil organic C cycling. Using rainfed olive farming as an experimental model, we determined the stability and accumulation levels of humic substances and applied these data to bacterial laccase-encoding gene expression and diversity in soils under four different agricultural management systems (bare soils under tillage/no tillage and vegetation cover under chemical/mechanical management).

Materials and Methods: Humic C (> 104 Da) was subjected to isoelectric focusing. The GC-MS method was used to analyze aromatic hydrocarbons. Real-Time PCR quantification and denaturing gradient gel electrophoresis (DGGE) for functional bacterial laccase-like multicopper oxidase (LMCO)-encoding genes and transcripts were also carried out.

Results: Soils under spontaneous vegetation, eliminated in springtime using mechanical methods for more than 30 years, showed the highest humic acid levels as well as the largest bacterial population rich in laccase genes and transcripts. The structure of the bacterial community based on LMCO genes also pointed to phylogenetic differences between these soils due to the impact of different management systems. Soils where herbicides were used to eliminate spontaneous vegetation once a year and those where pre-emergence herbicides resulted in bare soils clustered together for DNA-based DGGE analysis, which indicated a certain amount of microbial selection due to the application of herbicides. When LMCO-encoding gene expression was studied, soils where cover vegetation was managed either with herbicides or with mechanical methods showed less than 10% similarity, suggesting that the type of weed management strategy used can impact weed community composition and consequently laccase substrates derived from vegetation decay.

Conclusions: We suggest that the low humic acid content retrieved in the herbicide-treated soils was mainly related to the type (due to vegetal cover specialization) and smaller quantity (due to lower vegetal biomass levels) of phenolic substrates for laccase enzymes involved in humification processes. We also found that spontaneous vegetal cover managed using mechanical methods could be the best option for achieving C stabilization in rainfed Mediterranean agroecosystems.

Introduction

Due to their importance for human welfare, ecosystem services have, in recent years, been attracting increasing attention in the scientific literature. Although, traditionally, agroecosystems have been primarily regarded as service providers, their contribution to other types of ecosystem services have recently been seen as equally crucial (Millennium Ecosystem Assessment, 2005). Apart from providing food, forage and bioenergy, agricultural ecosystem services have other benefits such as water and climate system regulation, soil fertility and C sequestration (Chen et al., 2016). The latter involves the removal of C from the atmosphere and its storage in soils through physical and biological processes. As land-based biological C-mitigation strategies are regarded as important and viable pathways to climate stabilization (Canadell & Schulze, 2014), it is crucial to evaluate the effects of land-use changes on ecosystem functions. However, in semi-arid climates, agricultural soils are particularly vulnerable due to adverse environmental conditions and centuries-old farming systems (Metzger et al., 2006).

One of the principal agroecosystems in semi-arid Mediterranean areas is that used for perennial olive-orchard cropping. Olive is the most extensively cultivated fruit crop in the world, with more than 10 million hectares under cultivation in 2013 (FAOSTAT, 2015). Its economic and social importance is enormous, as reflected in the increasing per capita consumption of olive oil around the world. However, olive growing also has a considerable ecological impact in the Mediterranean region, where olive orchards constitute a fundamental part of the landscape. Nevertheless, these agroecosystems are particularly exposed to degradation because of low soil organic C (SOC) content.

The possibility of increasing the C-storage capacity of soils, with no significant reduction in yields, through the use of an appropriate soil management system, has been widely discussed (Scholes, Palm & Hickman, 2014). A recent meta-analysis concluded that the most appropriate strategy for maximizing SOC levels in the Mediterranean region is to combine organic inputs with reduced tillage management (Aguilera et al., 2013). However, in rainfed olive farming, SOC is mainly dependent on plant-derived C-breakdown (Castro et al., 2008; Moreno et al., 2009). SOC decomposition is generally thought to be regulated by biological and environmental conditions (Schmidt et al., 2011). Litter provides the carbon that supports microbial activity, returns part of the C to the atmosphere and stores the rest in the form of humic substances, a highly stable organic fraction considered to be the true agent responsible for soil C-sequestration mechanisms (Miralles et al., 2014). There is overwhelming evidence which shows that C storage in soils is mainly limited by the incorporation of new C rather than by the decomposition of existing C and also shows the critical role played by the soil microbial community in these processes (Lange et al., 2015). Thus, the microbial degradation of plant-waste lignin has been recognized as a C turnover bottleneck in soil (Theuerl & Buscot, 2010; Torres et al., 2014). The microbial breakdown of lignin has been well characterized, as have the enzymes involved, including laccase, lignin peroxidase and manganese peroxidase. However, the capacity to degrade lignin varies. By comparing the three classes of enzymes, Chen et al. (2011) have shown that laccases exhibit the most abundant aromatic-aromatic molecular interactions as well as the highest binding affinity. Laccases, generally laccase-like multicopper oxidases (LMCOs), whose kinetics have been widely studied, are probably the most common class of ligninolytic enzymes in soil. Although they have traditionally been regarded as fungal enzymes, there is mounting evidence which shows that bacterial LMCOs make a major contribution to lignin degradation and SOC turnover (Alexandre & Zhulin, 2000; Sinsabaug, 2010). Laccase-encoding genes have been successfully used in recent studies as molecular markers for SOC cycling under many different environmental conditions and demonstrate the known potential of bacteria to degrade lignin (Baldrian & López-Mondéjar, 2014). However, more studies are needed to elucidate the controversial role played by bacteria in soil C cycling (Kellner et al., 2008).

In this study, we identify the management practices which best help to increase SOC stocks in semi-arid agriculture in order to gain an insight into the contribution of soil bacteria to carbon sequestration in Mediterranean habitats. Using rain-fed olive farming as an experimental model, we thus determined the stability and accumulation levels of humic substances and then related these data to bacteria laccase-encoding gene expression and diversity levels in soils under four different agricultural management systems.

Materials and Methods

Experimental design

The main characteristics of the experiment and soils are described in Castro et al. (2008) and Moreno et al. (2009). The study was conducted in Jaen in south-east Spain (Supplemental Information 1), with a random block experimental plot design consisting of four treatments and four replicates. Each plot consisted of 16 olive trees, with the central four monitored trees surrounded by guard rows. The experiment, involving bare vs. covered soils, was begun in 1976 and was carried out over a period of 30 years. Samples were collected in May 2011. The treatments tested were: Tillage (T): Bare soils where weeds were eliminated by 3–4 annual passes with a disk harrow (at 30 cm deep) and/or a cultivator in spring, followed by a tine harrow in summer.

Non-tillage and no-cover (NC): Bare soils where weeds were eliminated by applying the pre-emergence herbicide oxyfluorfen in autumn. In the spring, glyphosate was applied locally.

Cover crop + herbicides (CH): Covered soils where weeds were left to grow each year to be eliminated in spring with herbicide glyphosate.

Cover crops + mower (CM): Covered soils where weeds were left to grow each year to be eliminated in spring with various passes of a chain mower.

Two samples were collected from the centre of each plot, using a modified soil-sample ring kit (Eijkelkamp), which consists of a 20 cm-deep sampling cylinder specifically manufactured for this purpose, and then bulked. To isolate the nucleic acids, subsamples of fresh soil were immediately frozen in liquid nitrogen. The samples were stored at −80 °C until molecular analyses were carried out.

Humic substances and isoelectric focusing

In this study, humic C was measured in sodium pyrophosphate extract (pH 7.1), a good extractant of humic compounds (Nannipieri et al., 1974; Piccolo et al., 1998). Humic substances were extracted at 37 °C for 24 h under shaking, using Na2P4O7 (0.1 M, pH 7.1) in a 1:10 w/v dry soil:extract ratio (Ceccanti et al., 1978). The suspension was then centrifuged at 12,400 g and filtered through a 0.22 μm Millipore membrane. The > 104 Da humic-derived fraction was obtained through ultrafiltration on the AMICON PM10 membrane of the extract. The C content of humic C (> 104 Da) was determined by acid digestion with 0.17 M K2Cr2O7 and concentrated (97%) H2SO4 at 140 °C for 2 h. A spectrophotometric method was used to quantify the Cr3+ produced by the reduction of Cr6+ (λ = 590 nm; Yeomans & Bremner, 1988).

Isoelectric focusing (IEF) of humic C (> 104 Da) was carried out in cylindrical gel rods (0.5 × 8 cm) containing polyacrylamide gel (5% w/v) and carrier ampholines in the pH 4–6 range (Bio-Rad Laboratories, Richmond, CA, USA) at a final concentration of 2% (Ceccanti et al., 1986; Ceccanti, Bonmati-Pont & Nannipieri, 1989). N,N,N′,N′-Tetramethyl-1,2-diaminomethane and ammoniumperoxy-disulfate were also added in a gel solution at 0.03%. At the top of the gel rod (cathode), 100 μl of humic C (> 104 Da) at 4.4% of glycerine were applied. A small amount (5 μl) of glycerine at 2.2% was added to the top of the sample to avoid mixing with the cathodic solution (NaOH 0.02 N); 0.01 M H3PO4 was used for the anodic cell. A pre-run of 1 h at the same current intensity and voltage used for the samples run was carried out for each gel tube (1.5 mA for each tube, 100–800 V); the samples run was then carried out for 2 h or longer until stable IEF banding was reached. The electrophoretic bands were scanned using a Bio-Rad GS 80 densitometer, giving a typical IEF profile for each soil investigated. The IEF peak area was determined for each soil IEF profile, assuming as 100% the area under the entire IEF profiles (representative of the total loaded C). Gel pH was measured at 0.5 cm intervals with an Orion microprocessor (model 901, Orion research) connected to a microelectrode gel-pHiler (Bio-Rad Laboratories, Richmond, CA, USA).

Aromatic hydrocarbons

Aromatic hydrocarbons were extracted ultrasonically from 5 g of freeze-dried soils using 10 ml of hexane/acetone (1:1 v/v). The 50 μl of C3-naphthalene in dichloromethane was added as a GC internal standard. After centrifugation and concentration, the solvent was dried by passing it over anhydrous sodium sulfate. The aromatic hydrocarbons were analyzed using a large volume injection-gas chromatography mass spectrometry (LVI–GC–MS) method. GC analyses were conducted on a Varian 450-GC fitted with a 1079 injector in split/splitless mode, a CTC Analytics CombiPal refrigerated autosampler and a Varian 240 Ion Trap as a mass spectrometer detector. The samples were injected in large volume mode using the 1079 PTV injector. The 10 μl were injected with the split open (split ratio of 1/100) for 0.5 min to purge the solvent, the split was then closed for 3 min, with the injector temperature ramped up, and the split was finally opened again (split ratio of 1/50). The injector temperature started at 60 °C for 0.5 min and then ramped up by 100 °C/min to 280 °C for 8 min. A Thermo TG-5SILMS fused silica capillary column (30 m × 0.25 mm × 0.25 μm) was used. Temperature ramp: 70 °C (3.5 min) > 25 °C/min to 180 °C (10 min) > 4 °C/min to 300 °C (13 min). The 10 μl of the extracts were injected according to the LVI method. The carrier gas was He at 0.9 ml min−1. Electron-impact ionization and detection in full scan (50–500 m/z) mode were used. Structural and quantitative information were obtained by comparing the mass spectra and retention times of the samples and standards with the aid of MS Workstation 6.9.1 software.

Nucleic acid isolation from soils and cDNA synthesis

For each soil-sample replicate, the total community-DNA was separately extracted from four 1 g subsamples using the bead-beating method by following the manufacturer’s instructions for the MoBioUltraClean Soil DNA Isolation kit (MoBio Laboratories, Solana Beach, CA, USA). The extracts were pooled and further concentrated at 35 °C to a final volume of 20 μl using a Savant Speedvac® concentrator.

Total RNA was extracted from four 2 g subsamples of each replicate following the manufacturer’s instructions for the MoBio RNA PowerSoil Total RNA Isolation kit (MoBio Laboratories, Solana Beach, CA, USA). To remove residual DNA, DNase I enzyme was added using the RNase-Free DNase Set (Roche Applied Science, Penzberg, Germany) following the manufacturer’s instructions. The extracts were pooled and further concentrated at 35 °C to a final volume of 80 μl using a Savant Speedvac® concentrator. The cDNA was synthesized from 1–2 μg of total DNase-treated RNA using the Transcriptor High Fidelity cDNA Synthesis Kit according to the manufacturer’s instructions (Roche Applied Science, Penzberg, Germany). The synthesis reaction was carried out at 50 °C for 30 min. The concentration and quality of the final DNA/RNA/cDNA samples were checked with the aid of a Nanodrop® ND-100 spectrometer (Nanodrop Technologies, Wilmington, DE, USA).

Quantitative real-time PCR assays

Real-time PCR quantification of DNA/cDNA for bacterial functional LMCO-encoding genes and transcripts was based on amplification with primers Cu1AF (5′-ACM WCB GTY CAY TGG CAY GG-3′) and Cu2R (5′-G RCT GTG GTA CCA GAA NGT NCC-3′), designed by Kellner et al. (2008). Agarose-gel electrophoresis was carried out to confirm the specific PCR product, resulting in a single band of the expected size (approximately 144 bp), with primer sequences excluded. To determine whether the primers were specific for amplification and detection of bacterial LMCO genes and transcripts of the soils tested, an LMCO gene-clone library was established after PCR of soil DNA samples. A total of 46 sequences were analysed: 38 showed homology to LMCO-genes, 6 showed homology to the cloning vector, 2 were badly sequenced and 1 did not show homology to the LMCO-genes (Supplemental Information 2).

Each 21-μl PCR reaction contained between 4 and 5.5 ng of DNA or between 0.2 and 0.3 μg of cDNA, 10.5 μl 2× iQ SYBR Green Supermix (Bio-Rad, Munich, Germany) and 400 nM of each primer. For each DNA/cDNA extracted, real-time PCR experiments were carried out three times, with the threshold cycle (Ct) determined in triplicate. The real-time PCR program consisted of 2 min at 50 °C for carryover prevention, 5 min at 94 °C for enzyme activation, followed by 5 cycles consisting of 94 °C for 15 s, 57–54 °C for 15 s, while decreasing the temperature by 0.5 °C in every cycle, 72 °C for 30 s, and 30 cycles consisting of 94 °C for 15 s, 54 °C for 15 s and 72 °C for 30 s; the fluorescence signal was measured at the 72 °C step for both DNA and cDNA. The final step lasted 7 min at 72 °C. The PCR amplification procedure was checked using a heat dissociation protocol (from 70 to 100 °C) following the final PCR cycle. The LMCO-encoding gene copy number was quantified on an iQ5 thermocycler using iQ5-Cycler software (Bio-Rad, Munich, Germany).

A standard curve was generated by using a recombinant plasmid containing one copy of the LMCO gene from soil bacteria. The curves were drawn according to the method described by Moreno et al. (2013). The relationship between the threshold cycle (Ct) and target-gene copy number and the copy numbers of the real-time standard was calculated as described by Qian et al. (2007) using the formula Ct = −3.089 × log10 (LMCO) + 30.254, R2 = 0.984. Target molecules were linear from 104 to 109 copies.

The potential presence of qPCR inhibitors was tested by mixing 1 μl (4–8 ng) of soils DNA extracts or 1 μl (50–350 ng) of soils cDNA extracts with a known amount of recombinant plasmid DNA (pCR®2.1, Invitrogen, Carlsbad, CA, USA) with the appropriate primers. Controls, where DNA and cDNA templates were replaced by filter-sterilized milliQ water, were carried out simultaneously. Ct values did not significantly differ between the DNA/cDNA extracts and the controls.

PCR-DGGE analyses

The community structure of bacterial LMCO-encoding genes and transcripts was studied by DGGE analysis, for which Cu1AF-DGGE and Cu2R primers were used. Primer Cu1AF-DGGE contained the same sequence as Cu1AF but with an additional 40-nucleotide GC-rich sequence (GC clamp) at the 5′ end of the primer.

DGGE analyses were conducted using 550 ng of the final PCR product loaded onto a 45–60% urea-formamide–polyacrylamide gel. An INGENYphorU System (Ingeny International BV, The Netherlands) was run at 200 V for 10 min followed by 75 V for 18 h at 58 °C in order to separate the fragments. Gels were silver stained with the Bio-Rad Silver Stain according to the standard DNA-staining protocol and photographed under UV light (λ = 254 nm) using a UVItec Gel Documentation system (UVItec Limited, Cambridge, UK).

Data analyses

Results are the means of 12 replicates (three per plot) for chemical analyses. With regard to nucleic acids, four extractions of each sample were carried out and then grouped together to obtain a single sample per plot. The results were subjected to a factorial analysis of variance (ANOVA) using the STATISTICA software program (StatSoft Inc., Tulsa, OK, USA). The post hoc Tukey HSD test in a one-way ANOVA was used. P-values of under 0.05 were considered to be evidence of statistical significance.

Band patterns in different DGGE lanes were compared with the aid of UVImap Analysis software (UVItec Limited, Cambridge, UK). DGGE banding data were used to estimate the Raup & Crick (1979) probability-based index of similarity (SRC). Raup and Crick’s (1979) probability method was used to determine whether similarities within and between samples were stronger or weaker than the randomized prediction. For this purpose, band-matching data were stored as a binary matrix and analyzed using the aforementioned probability-based index. The SRC index measures the probability of the randomized similarity being greater than or equal to the similarity actually observed; SRC values of over 0.95 or under 0.05 indicate similarity or dissimilarity, respectively, which are not random assortments of the same species (bands or OTUs) (Rowan et al., 2003). SRC and cluster analyses were carried out using the PAST (Paleontological Statistics) software program version 1.82b (Hammer, Harper & Ryan, 2001).

Results

Organic carbon fractions

We found that the high-molecular-weight (C > 104 Da) humic acid (HA) fractions behaved in a significantly different way from total SOC (Table 1). HA levels were much higher in CM, comparable in CH and NC and considerably lower in the T treatment. Figure 1 shows the IEF patterns of HAs. All the treatments showed the highest optical-density values (IEF peaks) in the pH range 4.5–4.2 (bands 1 and 2).

Table 1 Organic carbon fractions for tillage (T), non-tillage and no-cover (NC), cover vegetation + herbicides (CH), and cover vegetation + mower (CM) treatments (means ± SE).

For each parameter, significant differences are indicated by different letters (P < 0.05, ANOVA, Tukey post-hoc).

	C vegetal biomass (kg ha−1 year−1)*	SOC (g kg−1)**	HAs (C > 104 Da) (mg kg−1)	
T	3743 a	9.4 ± 3.1 a	306 ± 1.1 c	
NC	–	4.6 ± 0.7 b	390 ± 2.9 b	
CH	2617 a	6.8 ± 3.2 ab	443 ± 29 b	
CM	4059 a	8.3 ± 2.0 a	914 ± 6.6 a	
Notes:

SOC, Soil Organic Carbon; HAs, Humic Acids.

* From Castro et al. (2008).

** From Cañizares, Moreno & Benitez (2012).

Figure 1 Isoelectric focusing profiles of HAs from tillage (T), non-tillage and no-cover (NC), cover vegetation + herbicides (CH), and cover vegetation + mower (CM) soils.

Aromatic hydrocarbons

The results of the analysis of major aromatic hydrocarbons are shown in Table 2. Three main polycyclic aromatic hydrocarbons were detected in soil samples. 2,6-diisopropylnaphthalene (m/z = 239 + 199) was found to be present in all soils, although values were close to the detection limit in the CM treatment. The levels of pp-DDA, a polar metabolite of DDT also found in all soils, were significantly higher in the NC treatment. The third compound, present only in the NC and CH soils, was a mixture of four aromatic hydrocarbons (m/z = 265 + 266 + 331), whose mass spectra did not match any compounds in the library. To gain a better understanding of this mixture, its mass spectra are shown as supplementary material (Supplemental Information 3). The data appear to suggest that the C and D compounds, containing an odd number of N atoms, are dimers of A and B, respectively.

Table 2 Main aromatic hydrocarbons in tillage (T), non-tillage and no-cover (NC), cover vegetation + herbicides (CH), and cover vegetation + mower (CM) soils (means ± SE).

For each parameter, significant differences are indicated by different letters (P < 0.05, ANOVA, Tukey post-hoc).

	CAS 24157-81-1 (239 + 199)	p,p-DDA	A + B + C + D (265 + 266 + 331)	
T	0.22 ± 0.13 a	0.007 ± 0.005 b	nd	
NC	0.19 ± 0.05 a	0.021 ± 0.001 a	0.36 ± 0.05 a	
CH	0.25 ± 0.01 a	0.014 ± 0.004 b	0.22 ± 0.05 a	
CM	0.04 ± 0.03 b	0.011 ± 0.001 b	nd	
Note:

nd, not detected.

Real-time PCR assays and LMCO gene expression

Table 3 shows LMCO-encoding gene and transcript-copy numbers in the four treatments, both of which were much higher in CM soil and comparable in the other treatments.

Table 3 LMCO-encoding gene and transcript copy numbers for tillage (T), non-tillage and no-cover (NC), cover vegetation + herbicides (CH), and cover vegetation + mower (CM) soils (means ± SE).

For each parameter, significant differences are indicated by different letters (P < 0.05, ANOVA, Tukey post-hoc).

	LMCO gene copies g−1	LMCO transcript copies g−1	
T	9.79 × 104 ± 5.74 × 103 b	1.34 × 103 ± 3.16 × 102 b	
NC	1.02 × 105 ± 8.64 × 103 b	3.02 × 103 ± 8.38 × 102 b	
CH	7.89 × 104 ± 1.10 × 104 b	6.67 × 103 ± 3.21 × 103 b	
CM	5.71 × 105 ± 4.03 × 104 a	4.12 × 104 ± 4.86 × 103 a	

LMCO-encoding gene and transcript diversity

Figure 2 shows the DNA- and RNA-based DGGE dendrograms generated by Raup & Crick (1979) cluster analyses (see Supplemental Information 4). The SRC values, obtained by comparing the four treatments, are also summarized in Fig. 2. With respect to LMCO-encoding genes (Fig. 2A), T and NC soils differed significantly (SRC < 0.05). For the other pairs of DNA-DGGE profiles compared, the similarity was no greater than that the random prediction (0.95 > SRC > 0.05). However, SRC values comparing CM and the other treatments were very close to the dissimilarity boundary (SRC < 0.05), as evidenced by the solitary cluster where these soils were located.

Figure 2 Raup & Crick (1979) probability-based index of similarity cluster analyses and similarity values (SRC) between samples for profiles of DNA.

(A) and RNA-based DGGE analysis of LCMO-encoding genes (B) for tillage (T), non-tillage and no-cover (NC), cover vegetation + herbicides (CH), and cover vegetation + mower (CM) soils.

When the LMCO transcripts were evaluated (Fig. 2B), T and NC soils clustered together, showing a high degree of similarity (SRC ≈ 1). CH and CM soils showed low similarity (< 10%). After comparing CM with the other soils, the similarity was found to be no greater than that randomly predicted (0.95 > SRC > 0.05).

Discussion

In recent years, scientists have been discussing the relationship between agricultural management practices and soil-C sequestration. The long-term impact of several types of agricultural management on SOC dynamics has been extensively evaluated, some of which have been proposed as a means of offsetting anthropogenic CO2 emissions (Söderström et al., 2014). In particular, the advantage of including cover crops in cropping systems, as opposed to other management practices that increase SOC, has been highlighted (Poeplau & Don, 2015).

Data on SOC dynamics obtained in our previous studies suggest that no absolute advantages in terms of C sequestration appear to be associated with any of the four types of agricultural management tested. A previous paper reported comparable amounts of aboveground vegetal biomass per year in T, CH and CM treatments, as well as highly variable levels of annual biomass production which is typically found in semi-arid Mediterranean habitats, while leaves and root exudates from olive trees were the only C biomass input in the NC soil (see C vegetal biomass in Table 1 from Castro et al., 2008). SOC, measured after 30 years of experiments, was shown to be significantly lower in NC as compared to the T and CM treatments, though similar (P < 0.05) to the levels found in CH soils. However, no significant differences were observed between the former and latter treatments (SOC in Table 1 in Cañizares, Moreno & Benitez, 2012). Nevertheless, the outcomes for HAs have differed significantly, given that C-sequestration mechanisms are primarily based on the stabilization of C > 104 Da (Miralles et al., 2014). HA levels were much higher in CM soils, where spontaneous vegetation was allowed to grow each year and then eliminated in spring using mechanical methods. The IEF profiles confirm the stability levels of the humic substances isolated: larger quantities of organic C compounds were found at higher pH values (4.5–4.2) in the IEF gel, thus providing overwhelming evidence that the higher the isoelectric point, the more humified the organic matter (Ceccanti & Nannipieri, 1979; Govi, Ciavatta & Gessa, 1994).

These results provide supporting evidence that reduced tillage and green manure incorporation could represent the best option for C input and stabilization in the Mediterranean region (Álvaro-Fuentes et al., 2009; Aguilera et al., 2013; Garcia-Ruiz & Gomez-Muñoz, 2014; Garcia-Franco et al., 2015). This has usually been explained by the greater physical-chemical protection of the organic matter by microaggregates that this type of management provides. Tillage disrupts a larger proportion of aggregates, thus releasing higher levels of organic C as compared to no-tillage treatments (Six, Elliott & Paustian, 1999), which could explain why the T treatment exhibited the lowest HA levels. However, much of the current debate still revolves around SOC stabilization mechanisms, with existing evidence pointing to the synthesis of polyphenols as an essential step in HA formation. Polyphenols could be derived from lignin-degradation products or from microbial resynthesis, reflecting the importance of the specific type of vegetation and the critical role played by microbial ligninolytic enzymes in the soil, specifically phenoloxidase, laccase and peroxidase activities (Stevenson, 1994). Laccases are probably the largest class of ligninolytic enzymes in soil and perform different oxidative and polymerative functions. The enzymes of the former group are mainly involved in lignin breakdown, while the latter are chiefly involved in polymerizing soluble phenols, thereby contributing to humification (Gianfreda, Xu & Bollag, 1999). The data gathered in the present study suggest that there is a relationship between the number and expression of bacterial LMCO genes on the one hand and the quantity and stability of HAs on the other. Soils under vegetal cover managed by mechanical methods, where the highest HA levels were retrieved after 30 years of experiments, showed the largest bacterial population rich in laccase genes. Also, the environmental conditions facilitated a corresponding higher level of gene expression in these soils as compared to other treatments. The structure of the bacterial community based on LMCO genes also points to a phylogenetic difference in CM soils due to the management system used. CH and NC soils clustered together in the DNA-based DGGE analysis, suggesting a certain amount of microbial selection due to the application of herbicides. With respect to LMCO-encoding gene expression, CH and CM soils showed low similarity (< 10%), indicating that the type of weed management strategy used can impact weed populations and consequently laccase substrates derived from vegetation decay. Indeed, differences in weed community composition and the weed seed bank have previously been highlighted following long-term applications of preemergence herbicides as compared to non-chemical weed control management (Sosnoskie, Herms & Cardina, 2006; Barroso et al., 2015; Fracchiolla et al., 2015).

However, the results above did not provide confirmatory evidence concerning the role played by bacterial laccase in HA formation. Laccase substrates are quite diverse and extracellular bacterial laccase can oxidize other compounds such as dyes, pesticides and polycyclic aromatic hydrocarbons (Majeau, Brar & Tyagi, 2010). Nevertheless, in this study, almost all predominant aromatic hydrocarbon and herbicide metabolites were retrieved when herbicides were part of the agricultural management system; neither the type nor amount correlated with LMCO gene copies, transcript numbers or diversity.

Bringing together the above results, we found that management systems involving the application of herbicides could influence bacterial laccase activity and diversity either directly through their effect on soil microorganisms or indirectly through vegetal cover specialization. The literature provides considerable evidence to show that different types of lignin phenols are found in specific types of vegetation (Radosevich, Holt & Ghersa, 1997; Caseley, Cussans & Atkin, 2013), although there is no consensus on the effects of herbicides on microbial community structure and activity in soil (Jacobsen & Hjelmsø, 2014). Many authors have described the side-effects of glyphosate on microorganisms as of “little ecological significance” (Wardle & Parkinson, 1990), while others have contended that it has a considerable ecological impact (Nannipieri et al., 2003; Zobiole et al., 2011). In this long-term experiment, the claim that herbicides have no significant effect on soil bacteria after 30 years of herbicide applications is supported by previous data; soils under vegetation cover showed similar bacterial biomass and activity levels and characteristics whether spontaneous weeds were eliminated by mechanical or by chemical methods (Cañizares, Moreno & Benitez, 2012). However, apart from a different plant community pattern, herbicides could produce a collateral effect on certain bacterial activities related to the soil C cycle. Weed biomass and leaf/root exudates from olive trees were the sole C biomass input (Moreno et al., 2013), while litter-derived C was incorporated into both more and less chemically stable compounds. However, after 30 years of experiments, we detected an unexpected lower humification rate in CH as compared to CM soils. After reviewing the history of CH soils, we found that the herbicides diquat and paraquat had been applied over the years until their prohibition (Moreno et al., 2009) and were later replaced by glyphosate. On the basis of their mode of action, bipyridylium herbicides act by inhibiting photosynthesis (Dodge, 1971), while glyphosate inhibits the formation of phenolic precursors necessary for the production of lignin (Franz, Mao & Sikorski, 1997). We can thus assume that the lignin content of weeds incorporated into the soil in both circumstances was lower or simply different from that incorporated into the CM treatment, and it is also generally accepted that the synthesis of polyphenols from lignin is a critical step in C stabilization. Moreover, CH soils, where vegetation cover was eliminated once a year with herbicides, and NC soils, where weeds were prevented from growing by the application of herbicides, presented comparable levels of HAs as well as similar bacterial laccase gene-copy numbers, diversity and expression.

Conclusions

A long-term experiment comparing four types of agricultural management was conducted in a semi-arid Mediterranean agroecosystem. After 30 years of experiments, we found comparable stable levels of C in both non-tillage + no cover soils and soils under spontaneous vegetation cover managed using chemical methods. We suggest that the lower humic acid content retrieved in the herbicide-treated soils was mainly related to the type, due to vegetal cover specialization, and smaller quantity, due to lower vegetal biomass levels, of phenolic substrates for laccase enzymes involved in humification processes. We also conclude that spontaneous vegetation cover managed with mechanical methods is the best option to achieve C stabilization.

Supplemental Information

Supplemental Information 1 Sampling locations.

Click here for additional data file.

Supplemental Information 2 Mass spectra of the mixture of aromatic hydrocarbons (m/z = 265 + 266 + 331).

Click here for additional data file.

Supplemental Information 3 Raw data.

Checking the homology of Cu1AF-Cu2R primers to anneal and amplify the target DNA.

Click here for additional data file.

Supplemental Information 4 DGGE bands.

Computer-generated DGGE image used by the UVItec Gel Documentation software for analysing the bands.

Click here for additional data file.

The chromatographic analyses were carried out at the Scientific Instrumentation Service, Estación Experimental del Zaidín, CSIC, Granada, Spain. We wish to thank Serena Doni from ISE-CNR (Pisa, Italy) for the IEF analyses. We would also like to thank David Nesbitt and Michael O’Shea for editing and proofreading the latest English version of the manuscript.

Additional Information and Declarations

Competing Interests

Author Contributions

Data Deposition

The authors declare that they have no competing interests.

Beatriz Moreno conceived and designed the experiments, performed the experiments, analyzed the data, contributed reagents/materials/analysis tools, wrote the paper, prepared figures and/or tables, reviewed drafts of the paper.

Emilio Benitez conceived and designed the experiments, performed the experiments, analyzed the data, contributed reagents/materials/analysis tools, wrote the paper, prepared figures and/or tables, reviewed drafts of the paper.

The following information was supplied regarding data availability:

The raw data has been supplied as Supplemental Dataset Files.

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
