# Peer review of "Impact of agricultural management on bacterial laccase-encoding genes with possible implications for soil carbon storage in semi-arid Mediterranean olive farming"

_PeerJ, doi:10.7717/peerj.2257_

## Round 0.1 · original submission · Major Revisions

There are significant issues with the basic reporting of methods and results. As noted by the reviewers, the descriptions of the experimental treatments are confusing and should be clarified. See the recommendations from the reviewers on the overall organization of the manuscript.

Please do not make unsupported statements. For instance at L331 either include the data or, if it has been published elsewhere, provide a reference.

You should address the reviewers' comments about the experimental design, in particular the choice of substrates and enzymes examined, and details of the nucleic acid sampling, extraction and analysis. It is not clear where samples were pooled or where replicates were tested. If the latter, indicate the between replicate error. Also, address in detail the questions raised by Reviewer 2 on the validity of your findings.

There are numerous problems with English. Please correct all the specific examples noted below and in the reviewers' comments and have the manuscript read and edited by a native or fluent English speaker in order to catch any other errors.

L30 Consider replacing "dedication" with "importance"
L114 You do not specify the concentrations of your reagents
L125 There seems to be a word missing between "were" and "for"
L140 was the injector used in split mode or splitless? If the latter, what was the split ratio?
L145 Do not forget to superscript "-1"
L286 Check number agreement. Either "impact...has" or "impacts...have"
Numerous minor problems with English, e.g. L344, 353, 366, 376, 377

Reviewer 1 ·

Basic reporting

No Comments

Experimental design

No Comments

Validity of the findings

No Comments

Additional comments

In this paper from PeerJ Journal (#9546), the authors investigated the impact of four contrasting agricultural managements on soil properties in a semi-arid Mediterranean agroecosystem. The experiments design and experimental data analysis are adequate, and the experimental results and discussions are full of details. This is an interesting study and valuable to the field of soil carbon studies. The topic of the paper fits the scope of the journal. I would recommend this paper be accepted for publication after the following minor
comments are addressed:


1. There are many grammar errors in the manuscript. For example, p1, line21, "founded" should be "found". Such errors can be found in the whole manuscript.
2. The abstract needed to be re-written. This section is like a "materials and methods" section. Please keep in mind that abstract is a section focusing on the main results and findings.
3. p2, line 31, what are the "interest groups"?
4. Too many grammar errors. Please ask a native English speaker to help correct these errors.
5. p2, line 33-35, " including regulation of water and climate systems, soil fertility or C sequestration (Power et al., 2010).”About climate change on ecosystem, I suggest you cite the latest paper: Impacts of human activity modes and climate on heavy metal “spread” in groundwater are biased[J]. Chemosphere, 2016, 152: 439-445.
5. p4, line 90, "30 years"? what is the start date? What is the end date?
6. Why did you only measure the Humic substances and Aromatic hydrocarbons? Other substrates are also related to C cycle? To provide more background information on C, N and P cycles, I suggest you cite the reference: Global Landscape of Total Organic Carbon, Nitrogen and Phosphorus in Lake Water. Sci Rep 5, 15043 (2015).
7. There are many enzymes that have similar functions in soils. Why did you only concern laccase-like multicopper oxidases? I suggested that you cited more references to answer the question: (1). Understanding lignin-degrading reactions of ligninolytic enzymes: binding affinity and interactional profile." PloS one (2011): e25647; (2). Molecular basis of laccase bound to lignin: insight from comparative studies on the interaction of Trametes versicolor laccase with various lignin model compounds[J]. RSC Advances, 2015, 5(65): 52307-52313. (3) Bioremediation of soils contaminated with polycyclic aromatichydrocarbons, petroleum, pesticides, chlorophenols and heavy metals by composting: Applications, microbes and future research needs. Biotechnol Adv (2015), 33:745-755.
8. p16, line 375, " We hypothesize that the low humic acids pool retrieved in...". what are the basis for this hypothesis?

Reviewer 2 ·

Basic reporting

The introduction starts very general. Maybe the beginning can be shortened in order to add more information on effects of the agricultural managements used in this study, previously published results from this study site or diversity of LMCO genes (e.g. Kellner et al., 2008).

Although the treatments were explained in other publications before, authors should describe them at least one time clearly in the Material & Methods section (l.91-98) and use this description consistently throughout the manuscript. I found the abbreviation of the treatments difficult to understand and to keep in mind in which factors the treatments differ. For instance, in l.91 the Tillage explanation should include the information “bare soil” to contrast it with CH and CM (cover crop treatments). L.95 - Is treatment CH managed under reduced tillage as stated later in l.303? Maybe a small table would be good to indicate soil management, plant cover and weed elimination strategy for each treatment. In this respect, it is also difficult to understand l.21-23 in the abstract: you should make clear that both treatments received herbicides. It would be also helpful to add the abbreviations of the treatment in parentheses, e.g. l.366 CH and l. 367 NC

l.236-243 I did not understand to which “previous” paper you refer to (l.236). I recommend to move this paragraph to the discussion section which is a more appropriate place to relate your results (HA content) to the results of Castro et al. 2008/Canizares et al. 2012 (Table 1).

Experimental design

l.99-101 Did you sample two bulk soil samples per plot which were then mixed? Was the chemical analysis (humic substances, aromatic hydrocarbons etc) done per replicate (4 per treatment)? Please add that information to Material & Methods section. Later at l.216, you stated that “results are the means of 12 replicates (three per plot)”, but I was wondering from where they came from. For DNA/RNA extraction you stated that replicates were processed separately. Which DNA/RNA extracts did you pool- from 4 subsamples per replicate (l.154+l.160)? In this respect, I was wondering why only one DGGE fingerprint per treatment is shown (supplement) and not of the 4 four independent replicates? Is it a mixture of the 4 replicate DNAs? However, in lines 280-281 you state that there was a high similarity between replicates indicating you did a separate DGGE and cluster analysis for them?!

l.181 Why did you include the 5-cycles step with a stepwise decrease of the annealing temperature? l.183 “in both cases” is referring to quantification of LMCO-genes from DNA/cDNA?

Validity of the findings

l.265/Table 3- I recommend to relate LMCO gene copy numbers to number of bacterial 16S rRNA genes since conclusions for CM treatment could be biased due to previously reported higher bacterial biomass and 16S rRNA gene copy numbers in this treatment (Moreno et al. 2009).

Regarding LMCO gene/transcript copy numbers (Table 3) and the standard used for quantification (l.192), numbers were on the lower range of detection. I was wondering whether you performed a nested PCR for DGGE analysis in order to enrich for those functional genes.

I think it would be really interesting to get more insights into diversity of LMCO-genes in those soils by construction of a clone library. This would also greatly help to provide some background to the discussion l.328-332 which is at the moment very speculative. You should also consider here that separate clustering of transcripts from CH and CM soils (l.329) was not significant.

I was wondering why you decided to use the SRC of Raup & Crick. Is it suitable for DGGE data since it was developed for measurement of faunal similarity based only on presence/absence not taking into account the intensity of DGGE band? I would prefer to have the original DGGE fingerprints provided in the supplement without band matching. In this respect, why were the upper DGGE bands not considered for band matching?

Additional comments

Minor comments: Please refer to Tables/Figures that were provided in the supplement. How does the sequence analysis/Homology file relate to the present manuscript?
l.18 explain the first time use of abbreviation “DGGE”
l.21 founded should be changed to found (check throughout the manuscript)
l.23-26 Sentence is difficult to understand. Please consider rephrasing.
l.34 change “including” to “such as”
l.39 reference: Schulze
l.84 delete “layout”
l.107 humic C was measured
l.108 check space “. Humic”
l.120 check space “(cathode), 100 µl”.
In general check spaces before units (e.g. l.137, l.144)
l.125 explain abbreviation IEF and check the sentence (word missing?)
l.136 Please revise: soils were subjected to ultrasonic extraction … OR: Aromatic hydrocarbons were extracted ultrasonically from 5 g of soil…
l.151 total community-DNA
l.162 What do you mean with “RNA-DNase”?
l.170/171 Real-time PCR quantification … was based on amplification with primers CU1AF …
l.170 please include “bacterial” functional LMCO-encoding genes
l.186 please revise “ Quantification of the LMCO-encoding gene copy number”
l.194-195 please replace “sediments” by “soil”?
l.203-204 Please revise this sentence: the community structure of bacterial LMCO-encoding genes and transcripts was studied by DGGE analysis
l.208/209 The Netherlands
l.255 What do you mean with “the same occurred for pp-DDA”?
l.277 I suggest the LMCO-structure (transcripts) between CH and CM is not significantly dissimilar (SRC = 0.174). Please check!
l.292 Please cite the previous studies here and refer to Table 1
l.333 I recommend to revise this sentence since you did not test this.
l.348-352 I don’t agree with this sentence: Moreno et al. 2009 found a lower microbial functional diversity due to herbicide application; is reference Canizares et al. 2012 correct here?
l.368 gene-copy numbers
l.374 non-tillage

According to PeerJ standards, the funding source can be deleted in the Acknowledgment section since it will be stated separately.

Check in-text citation rules of PeerJ (“For three or fewer authors, list all author names (e.g. Smith, Jones & Johnson, 2004”)

l.395 I couldn’t find the citation Alexandre & Zhulin 2000 in the manuscript
l.414 olive-grove management systems
l.437 Senesi, N.
l.440 software package
l.514 soybeans . Journal of Applied Microbiology
Table 3 legend: Instead of „molecular estimates“ state „LMCO-encoding gene and transcript copy numbers“

---

## Round 0.2 · Minor Revisions

Since the data for weed community composition is unavailable, please delete the reference to "Data not shown" in lines 301-302 unless it has now been published and you are able to cite it. This should not adversely affect your discussion because you have cited additional supporting work.

In addition, please also address the individual points raised by Reviewer 2.

Reviewer 1 ·

Basic reporting

The authors have successfully addressed my concerns.

Experimental design

The authors have successfully addressed my concerns.

Validity of the findings

The authors have successfully addressed my concerns.

Additional comments

The authors have successfully addressed my concerns.

Reviewer 2 ·

Basic reporting

no comments

Experimental design

no comments

Validity of the findings

no comments

Additional comments

The manuscript by Moreno & Benitez improved after revision. Most of the points raised by the Editor and reviewers were adequately answered and the manuscript was revised if applicable. Before publication, however, I recommend to check the following comments again:
Abstract: The abstract was changed following suggestions of Reviewer 1. The current abstract, however, starts now with introduction of laccases which is somehow misleading since the focus of the study is the SOC content in agricultural, Mediterranean soils and the impact of agricultural management. I recommend to make the rationale and hypothesis of your study, that is stated at l.92-94, more clear in the abstract (instead of writing that “further research in this field is necessary”).
Materials & Methods:
l.108 The start and end date of the long-term experiment was added in order to answer the question of reviewer 1. Do I understand correctly that samples were taken in 2011 when the long-term management of soils stopped already (year 2006)? If yes, I think it is important to mention this in the discussion because it suggests that this management effect persists in soil. Or was management carried out for more than 30 years (until 2011)?
Results:
l. 299-300 I recommend to add here the sentence from discussion “CH and CM soils showed low similarity” (l.355-356)

Minor remarks:
l.22 delete “of DNA/RNA”
l.23 delete “laccase like multicopper oxidase” and use only abbreviation (introduced before)
l.24-26 I suggest to revise this sentence “ Soils under spontaneous vegetation, eliminated… for more than 30 years, showed the highest humic acid levels …”
l.34 I suggest to revise also this sentence “…, suggesting the formation of different laccase substrates from vegetation cover decay when herbicides are used.” (and also in l.356-357)
l.52 Schulze not Schulzel
l.137, 309, 377 check spaces before “et al.”
l.141 check space 0.02 N
l.205 I guess it should be 10.5 µl rather than 10.5 ml
l.231 instead of “subjected to” please write “studied by DGGE analysis”
l.240 UVItec
l.244 Were 12 replicates available for all analyses? Or only for chemical analyses + qPCR
l.275 please revise: I don’t understand what you refer to with “A similar trend was observed for pp-DDA” (Like compounds 293+299, pp-DDA was found in all soils, but a lower level was detected in T than in CM)
l.319 delete “also”
l.353 check space “used. CH”
l.448, 452,454,478,505, 543,552,554 Check spaces in references
l.488 Griffiths, R.I.
l.492 Brar, S.K.
l.511 podzol. Soil Biology
l.540 delete one time “doi”

---

## Round 0.3 · Minor Revisions

(1) The claim regarding the similarity between replicates, not just the words "data not shown" should be deleted if the replicate data is not available. This requires deletion of L304-5 in latest revision. I should perhaps have been clearer in my earlier decision letter.

(2) Typographical issues:
(i) Remove bold from "studied in" (L231) in final version
(ii) There are some inconsistencies in the reference section: punctuation between journal and volume, space/no space after "doi:", Journal names abbreviated/in full. Please ensure that these are checked and corrected before publication.

---

## Round 0.4 · accepted · Accept

Thank you for your prompt attention to the last round of minor revisions.